# Effects of 10-Day Complete Fasting on Physiological Homeostasis, Nutrition and Health Markers in Male Adults

**DOI:** 10.3390/nu14183860

**Published:** 2022-09-18

**Authors:** Zhongquan Dai, Hongyu Zhang, Feng Wu, Ying Chen, Chao Yang, Hailong Wang, Xiukun Sui, Yaxiu Guo, Bingmu Xin, Zhifeng Guo, Jianghui Xiong, Bin Wu, Yinghui Li

**Affiliations:** 1State Key Laboratory of Space Medicine Fundamentals and Application, China Astronaut Research and Training Center, No. 26, Beiqing Road, Haidian District, Beijing 100094, China; 2Engineering Research Center of Human Circadian Rhythm and Sleep, Space Science and Technology Institute (Shenzhen), Shenzhen 518117, China

**Keywords:** prolonged fasting, water-only fasting, metabolic homeostasis, therapeutic strategy, chronic diseases

## Abstract

Fasting shows great potential in preventing chronic diseases and has to be surmounted under some extraordinary circumstances. This study aimed to investigate the safety, time effects of metabolic homeostasis and health indexes during prolonged fasting. Thirteen participants were recruited to conduct a 10-day complete fasting (CF) in a controlled health research building under medical supervision including 3-day Baseline (BL), 10-day CF, 4-day calorie restriction (CR) and 5-day full recovery (FR). Body healthy status was assessed by surveying pulse, blood pressure, body weight (BW), blood glucose and ketones, body composition and nutritional and biochemistry indexes at different times. BW declined about 7.28 kg (−9.8%) after 10-day CF, accompanied by increased pulse and decreased systolic blood pressure, but there were no changes to the myocardial enzymogram. Body composition analysis showed fat mass was constantly lost, but lean mass could recover after CR. The energy substrate switch from glucose to ketone occurred and formed a stable dynamic balance between 3–6 days of CF. The lipid metabolism presented increased total cholesterol, LDL-C, ApoA1 and almost no changes to TG and HDL-C. Prolonged CF did not influence liver function, but induced a slight decrease of kidney function. The interesting results came from the marked increase of lipid-soluble vitamins and a significant decrease of sodium and chlorine. Adults could well tol-erate a 10-day CF. A new metabolic homeostasis was achieved. No vitamins but NaCl supplement should be considered. These findings provide evidence to design a new fasting strategy for clinical practice.

## 1. Introduction

Fasting has been practiced for millennia for religious, ethical or health reasons. Now, it is a popular therapeutic strategy to prevent chronic disease [1,2] and a huge problem during some extraordinary circumstances in which people may be unable to obtain anything edible for days. To endure prolonged fasting, the body undergoes important physiological adjustments to keep its function. However, whether severe fasting is safe or leads to a new energy metabolic homeostasis is largely unknown.

The changes in modern lifestyles and excessive food intake increase the risk of developing chronic diseases. The first therapeutic strategy to prevent these diseases is a lifestyle change such as dietary interventions and physical activity more than pharmacological treatments [3]. In humans, fasting is characterized by consciously eating no or minimal amounts of food and caloric beverages for periods that typically range from 12 h to weeks [4]. Emerging discoveries drive a growing public perception that fasting is beneficial for many aspects of human health and shows great potential for the therapy of metabolic diseases [5,6]. Although the prevalent fasting models such as intermittent fasting (IF), alternative day fasting (ADF) and time-restricted feeding (TRF) have been demonstrated to potentially prevent diseases and manage body fitness with minor side effects [4,7,8], the ameliorative effects of fasting show great diversities mainly due to individual differences and various short-term complete fasting (CF) models, which influence its large-scale application to an array of diseases [9,10,11,12]. Mechanistically, the frequently repeated long-term fasting intervals may favor preferential reduction of ectopic fat, and beneficially modulate aspects of glucose and lipid metabolism [13]. Prolonged fasting has been investigated in various clinical conditions for chronic inflammatory, obesity, hypertension and metabolic syndrome [14]. However, prolonged fasting may induce both favorable and harmful effects and the net benefit of pathways activated by fasting are context-dependent. The results of serial transcriptomics of subcutaneous adipose tissue and systematically circulating inflammatory markers from 10-day fasting show an unexpected dominant signal of inflammation [15]. Fasting for 60 h decreased insulin-mediated peripheral glucose uptake, which indicated that prolonged fasting induced profound peripheral insulin resistance [16,17]. Growing evidence shows the benefits of intermittent energy restriction on glucose and lipid homeostasis in the short-to-medium term. However, there is a lack of safe prolonged fasting protocols to guide physicians in its prescription [18] because there are few controlled trials of any form of fasting that gauge the effects [19]. More safety studies about long-term fasting are required [13,20].

It is well known that fasting results in ketogenesis and an energy substrate switch after the exhaustion of glycogen. Andriy et al. presumed that depletion of stored glycogen played major roles in the health benefits from IF [21]. The metabolic switch does not truly take effect during short-time fasting because the glycogen, free fatty acid and amino acid stored in the tissues is able to supply energy substrates for 24–36 h, depending on the physical activity and recovery intervals [22,23]. Prevalent IF and ADF refer to CF lengths from 16 to 48 h, and TRE is characterized by intake of food within 8 h per day, such as during Ramadan. Additionally, one difference caused by different fasting ways is the effect of various growth factors and metabolites. IF caused more frequent but less pronounced changes than prolonged fasting [4]. The major effects of fasting relevant to aging and diseases are the changes to IGF-1, glucose and insulin. Five days of fasting caused an over 60% decrease of IGF-1 [24], but chronic calorie restriction (CR) does not lead to a decline of IGF-I [25]. Ramadan fasting for 28 d had no major effects on body composition, or glucose metabolism in healthy lean males [26]. It is important to identify the lasting period and frequency of CF of the change of specific growth factors, and metabolic and nutritional markers in controlled trials [19].

Taken together, although ketone bodies, deposited fat and gluconeogenesis allow most human beings to survive 30 or more days in the absence of any food, there are still few human being experiments that explore the systematic changes of prolonged fasting on a healthy body. We proposed that the short time of CF is the main cause of different or individual benefits from fasting. Then, a 10-day CF human experiment was conducted under controlled conditions to systematically estimate its safety, tolerance and time effects on health, psychological mood and brain functions. Our previous reports have presented that 10-day CF affected subjective sensations but not cognitive abilities, which was associated with the energy substance switch in the middle of CF [27], and previous fasting experience was important to prevent negative feelings during prolonged fasting [28]. In the present, we minutely depicted the time effects of 10-day CF followed by refeeding on body health, physiological and nutritional status assessed by pulse, blood pressure, body weight (BW), blood glucose and ketones, body composition and nutritional and biochemistry indexes to evaluate health risks and the application advantage of long-term CF.

## 2. Materials and Methods

### 2.1. Participants

We performed a 10-day water-only fasting in male adults under medical supervision conditions with detailed descriptions in our previous publications [27,28]. In brief, 60 interested Chinese participants were invited for further screening and 44 participants dropped out due to age, weight, BMI, underlying health conditions or religious beliefs. Three dropped out for personal reasons before the start of the experiment. Finally, 13 male participants undertook water only ad libitum drinking during the fasting period in a controlled laboratory building, accompanied with a mild-intensity lifestyle program. The exclusion criteria included: history of eating disorders or any chronic illnesses, including diabetes mellitus, cancer, cardiovascular disease, metabolic diseases and tobacco or alcohol dependence, which were reviewed during a complete medical examination before the experiment.

### 2.2. Study Design

The study was approved by the Ethics Committee of the Space Science and Technology Institute (Shenzhen) (SISCJK201805001) and followed the ethical standards of the 1964 Declaration of Helsinki and its later amendments. All participants signed their written informed consent form and were informed by detailed experimental design and process of this study before the start of the experiment. A 22-day experiment was divided into four phases (3-day Baseline (BL), 10-day CF, 4-day CR and 5-day Full Recovery (FR). During the second phase, the participants were only permitted to drink Nongfu Spring water (including Ca ≥ 400, Mg ≥ 50, Na ≥ 80, K ≥ 35, metasilicate ≥ 180 µg/100 mL) ad libitum and performed normal ambulatory activities in a health research building. In the third phase, the participants were gradually given increased amounts of food to protect their digestive system after CF. During the final phase, the participants were allowed to return to their normal eating habit.

### 2.3. Physiological Parameters and Blood Chemistry

Resting blood pressure and pulse were measured immediately after getting up (about 6:30–7:30 am) without any eating, drinking and exercise every day using a blood pressure examination (OMRON HEM-7121, Dalian, China). Blood samples were collected at 7 am at the BF1 (1 day before fasting), CF3 (3rd day of CF), CF6, CF9, CR3 (3rd day of CR) and FR5 (5th day of FR), and urine samples were collected before 9 am at the same time point. Some samples were promptly sent to a certified hospital (Grade III Level A hospital), Longgang Central Hospital in Shenzhen, using the standard protocol to test the regular indexes. The others were centrifuged at 3000× *g* for 10 min and stored at −70 ℃ until assay. Some special indicators were measured by Beijing Zhongtong Lanbo Medical Laboratory Co., Ltd. (Beijing, China) with the most authoritative third-party testing inspection agency certification.

Blood glucose and β-hydroxybutyrate (BHB), the main form of ketone bodies, were measured with the test strip (FreeStyle Optium, Abbott, Alameda, CA, USA) everyday morning during CF and CR period until FR. A Homeostatic Model Assessment of Insulin Resistance (HOMA-IR) was calculated by the formulae: (fasting insulin in mIU/L × fasting glucose in mmol/L)/22.5. Glomerular filtration rate was estimated by CDK-EPI_2009src_ equation: 141 × (Scr/0.9)^α^ × 0.993^Age^, where α is −0.411 if Scr less than 0.9 mg/dL or is −1.209 if Scr more than 0.9 mg/dL. 

### 2.4. Anthropometric and Body Composition Measures

BW was measured using an electric balance (Mi Smart scales, Hefei, China) with an accuracy of 0.05 everyday morning. Height was metered with a stadiometer (HNH-219, OMRON, Shenzhen, China) during the physical examinations at Longgang Central Hospital in Shenzhen, China. BMI was determined according to the formula of BW/height^2^. The whole-body composition was assessed at the BF2, CF6 and FR5 by a dual X-ray absorptiometry (DEXA) scan (Discovery W, Hologic, Marlborough, MA, USA) and a BIO-impedance analyzer (SFB7, Impedimed Brisbane, Brisbane, Australia) according to the standard operational procedure. Participants wore lightweight clothing during this measurement. Fat mass index (FMI) and lean mass index (LMI) were calculated as fat mass or lean body mass (kg) divided by the square of height (m^2^).

### 2.5. Circumference and Skinfold Thickness Measurements

The circumference of the mid-upper arm (MUA), chest, waist and hip were manually gauged with the subject relaxed standing using a nonstrectable measuring tape after the DEXA scan and recorded in centimeters to the nearest millimeter. The MUA circumference was at the midpoint of the upper arm between the olecranon and acromion. The chest circumference was just below the level of the nipples in the expiratory stage, the waist circumference at the level of the umbilicus, and the hip circumference at the trochanter level, respectively. All measurements were repeated 4 times to obtain the average value.

The skinfold thickness (SFT) measurements were taken from the triceps, subscapularis and abdomen with a digital skinfold caliper (Deqing, China) at the same time point of the DEXA scan. All measurements were taken on the same side of the body before, during and after CF and repeated 4 times for each test. The triceps SFT was measured on the midline of the posterior aspect of the upper arm. The subscapular SFT was measured as a diagonal fold 1–2 cm from the inferior angle of the scapula. The abdomen SFT was measured at 5 cm on the right side of the umbilicus.

### 2.6. Resting Metabolic Rate Analysis

Resting metabolic rate (RMR) and respiratory quotient (RQ) were assessed by an indirect calorimetry (Cortex MetaMax 3B, Leipzig, German) for 20 min on the BF1/2, CF3/4, CF9/10, CR3/4 and FR4/5 during the prolonged fasting experiment. Prior to the test, the system was turned on for at least 20 min, then calibrated according to the manufacturer’s recommendations by a calibration kit (Cortex, Leipzig, German). The weight and height of participants were recorded as input information in the Metasoft system (Version 3.0, Cortex, Leipzig, Germany). On the examined morning, the participant imperturbably laid on a check bed immediately after getting up at room temperature (25.0 ± 1.0 °C). The mask covered the subject’s nose and mouth to perform breath-by-breath analyses of volume, O_2_ concentration, CO_2_ concentration by the volume meter, temperature and O_2_ and CO_2_ sensor. The oxygen volume consumed in liters per minute (VO_2_), and carbon dioxide volume expired in liters per minute (VCO_2_) were automatically and continuously calculated during the testing period by Metasoft and used to artificially reckon RMR with the Weir equation [29] (RMR = (3.9 × VO_2_ + 1.1 × VCO_2_) × 1.44 × 1000 (VO_2_: mL/min); VCO_2_: mL/min)) and RQ = VO_2_/VCO_2_ by Excel software.

### 2.7. Statistical Analysis

All statistical analysis was performed using GraphPad Prism(version 8.0 San Diego, CA, USA). To assess the overtime fasting effects on the physiological parameters, a repeated-measures one-way ANOVA was conducted after Shapiro–Wilk normality and lognormality tests. Data was presented as mean ± SD. The significance for all the statistical analyses was set at *p* < 0.05.

## 3. Results

In total, 13 male participants had a mean age of 39.6 ± 7.9 (range 28–55) years, mean weight of 72.1 ± 11.9 (range 54.2–92.2) kg, a mean body mass index of 24.6 ± 3.5 (range 19.2–31.9) and all finished the complete prolonged fasting experiment.

### 3.1. Body Weight

The BW of each participant presented a similar change trend (Figure 1). The average BW significantly decreased (7.28 ± 1.46 kg, −9.8% ± 0.01%, *p* < 0.001) after 10-day CF. BW was speedily lost in the initial 3 days with an average of −1.20 ± 0.63 kg per day compared with the previous day. Since then, BW descent speed progressively slowed down to 0.28 ± 0.42 kg on the CF10. The downtrend of BW lasted to the CR2. A significant increase occurred on the FR2, when compared with the CF10. The BW was still less 3.45 ± 2.10 kg than the BF after 4-day FR refeeding.

### 3.2. Blood Pressure, Pulse and Myocardial Enzymograms

As shown in Figure 2a, pulses gradually rose during CF, reached their highest level on the CF8 with an increase (36.4%), began to markedly descend on the CR3 (vs. CR2) and recovered to baseline level on the FR3. Nevertheless, systolic blood pressure (SBP) showed a decreasing trend with significant decline on the CF7 (−6.44%) and reached the lowest level on the FR1 (−14.68%), then recovered to normal on the FR3 compared with the BF (Figure 2b). Furthermore, the myocardial enzymograms of creatine kinase (CK), CK-MB, lactate dehydrogenase (LDH) and α-hydroxybutyric dehydrogenase (HBDH) showed no significant changes during CF and diet recovery (Figure 2c–f). However, a slight increase of CK and CK-MB was observed on the CF3, about 30% and 25%, respectively, but there were no significant differences.

### 3.3. Blood Routine Index and Nutrition Indicator

The blood routine indexes were examined during the fasting experiment. When compared with the BF, the MONO markedly increased about 27.88% (*p* = 0.038) on the CF3; and NEUT and NEUT% significantly decreased 25.52% (*p* = 0.045) and 10.86% (*p* = 0.049) on the CR3. There was a significant increase of RBC (13.19%, *p* = 0.0139), MONO% (21.30, *p* = 0.0421), HGB (8.48%, *p* = 0.0373) and HCT (9.45%, *p* = 0.006) on the CF9, as shown in Table 1. Moreover, the other 17 indexes did not show significant alterations at the six examined time points (Appendix A).

As shown in Figure 3a, the blood concentration of nutritional protein indicators, as total protein (TP), albumin (ALB) and globin (GLB) was always stable during the whole experiment, except for a decrease of TP on the FR5. However, when compared with the BF, the prealbumin showed a significant decline (−13.8%) on the CF3 and reached its minimal concentration on the CF9 (181.0 ± 26.34 mg/L, −39.6%). They remained at this low level until the FR5 with a decrease of 17.9% (Figure 3b). The change tendency of retinol binding protein (RBP) during the CF and CR period was the same as prealbumin with a difference of decreasing degree (Figure 3c). The transferrin also markedly decreased by 15.1% on the CF3 and subsequently remained at this level during the CR and FR period (Figure 3d). These short-life proteins were still lower on the FR5 than the BF.

### 3.4. Body Composition

For radiation safety protection, human body composition was only examined on the BF3 (3 days before fasting), CF6 and FR5 by DEXA. Fat mass (FM), lean mass (LM) and their percentage were analyzed as total or regional body. Compared with the BF3, total FM significantly decreased by about 10.7% and 17.2% on the CF6 and FR5, respectively (Figure 4a). However, total LM markedly decreased by 9.2% on the CF6 and recovered to the baseline level on the FR5 (Figure 4b). The percentage of total FM of the whole body only declined 2.0% on the CF6, but sustainedly decreased to 85.7% on the FR5 (Figure 4c). In contrast, the percentage of total LM to the whole body showed no change on the CF6, but markedly increased by 5.5% on the FR5 (Figure 4d). The ratio of LM/FM significantly increased by 4.2% and 23.9% on the CF6 and FR5, respectively. FMI markedly declined from 6.76 ± 2.28 on the BF3 to 6.03 ± 2.21 on the CF6 and 5.64 ± 2.01 on the FR5 (*p* < 0.001), but the LMI decreased from 16.70 ± 1.51 (BF3) to 15.15 ± 1.26 (CF6, *p* < 0.001), and recovered to 16.95 ± 1.19 on the FR5.

To understand the distribution shift of body composition, FM and LM or their percentage of different body compartments were also analyzed. Compared with the BF3, fat content in the limbs, trunk and head significantly decreased on the CF6 with an average decline of 10.9%, 11.1% and 5.4%, respectively. On the FR5, fat content in the limbs and trunk remained significant, with a decrease of 16.2% and 19.7%, respectively. Fat content in the head on the FR5 was lower than the BF3, but there was no significant difference (Figure 4e). There were distinct alterations of fat percentage in the limbs on the CF6 and FR5, in the trunk on the FR5 and in the head on the CF6. No changes of fat percentage occurred in the trunk on the CF6 and in the head on the FR5 (Figure 4c). LM in the limbs, trunk and head significantly declined on the CF6 with an average of 7.4%, 11.5% and 4.9%, respectively, which all recovered to baseline level on the FR5 (Figure 4f). The percentage of LM showed significant elevation in the limbs and a decrease in the head on the CF6. There was a significant rise of LM percentage in the limbs and trunk, but not in the head on the FR5 (Figure 4d). The FM ratio of trunk to whole body showed a sustained decrease from 0.530 ± 0.04 on the BF3, 0.527 ± 0.03 on the CF6 to 0.515 ± 0.03 (*p* < 0.001) on the FR5, which did not happen in the limbs (Figure 4g).

Using the standard protocol of the manufacturer, we analyzed FM and its percentage of android and gynoid compartments. After a 6-day CF, fat content in android and gynoid markedly declined by 14.7% and 7.9% (*p* < 0.001) respectively, which constantly decreased by 20.1% and 15.8% on the FR5 by comparing with the BF3. Compared with the CF6, FM of these two compartments also significantly decreased by 6.4% and 8.6% on the FR5 (Figure 4h). FM percentage of android and gynoid showed no significant alteration on the CF6, but notedly changed on the FR5 with a 15.4% and 12.1% decrease, respectively (Figure 4i). The android/gynoid FM ratio significantly decreased from 7.8% (0.63 ± 0.14 vs. 0.58 ± 0.13, *p* = 0.021) on the CF6, but no marked difference was noted on the FR5 (0.61 ± 0.12, *p* = 0.156). Fat free mass (lean + bone) of android and gynoid also significantly decreased by 13.1% and 6.5%, respectively, on the CF6, but no marked changes on occurred the FR5 compared with the BF3 (Figure 4j). Estimated visceral adipose tissue (Est.VAT) mass showed more decrease degree with 20.7% and 18.1% (*p* < 0.001) on the CF6 and FR5, respectively, as did the EST. VAT volume with a decrease of −20.0% and −17.3% (Figure 4k). EST.VAT mass and volume on the FR5 showed no difference with the CF6.

Total body water (TBW), extracellular fluid (ECF), intracellular fluid (ICF) and their percentage were examined by Bio-impedance analyzer SFB7 at the same time point of the DEXA scan. On the CF6, TBW significantly decreased by 11.4% and was associated with no significant change to TBW percentage (TBW%). On the FR5, TBW substantially recovered to prefasting levels and TBW% significantly increased by 3.7% (Figure 4i). In addition, ECF, ICF and their percentage on the CF6 markedly declined by 15.1%, 8.7%, 4.2% and 3.1%, respectively. After 5 days of FR, ECF and ICF substantially recovered to prefasting levels. ECF% showed a significant increase of 4.1%, but ICF% remained a significant decrease of 3.1%, compared with the BF3 (Figure 4l,m).

SFT is widely used to evaluate body composition too. On the CF6, SFT of three examined sites significantly decreased by 20.4% (triceps), 15.3% (subscapularis) and 17.1% (abdomen), respectively, and was continuously lower (17.1%, 11.1% and 17.0%) on the FR5 than that on the BF3 (Figure 4n). The circumferences of MUA, chest, waist and hips was measured at the same time. As expected, significant decreases of all four circumferences were observed on the CF6 at 3.7%, 2.8%, 6.4% and 2.7%, respectively. No recovery occurred in chest and MUA sites on the FR5, but circumferences of the waist and hips almost returned to baseline levels (Figure 4o).

### 3.5. Glycometabolism and Lipometabolism

The concentration of blood glucose and BHB was recorded everyday by a strip test until the CR4, when their levels fully recovered to the baseline. As expected, glucose showed a sustained significant decrease from 4.96 ± 0.44 mmol/L of the BF to 4.18 ± 0.54 mmol/L of the CF1 and reached its lowest at 3.508 ± 0.397 mmol/L on the CF4. From CF8, blood glucose showed a slight increase and was maintained at this level in the following fasting days and refeeding period. Contrarily, BHB continuously rose from the CF1 (1.308 ± 1.053 vs. 0.177 ± 0.044 mmol/L of the BF) and ran up to its highest on the CF5 and maintained at this level until the CF9. Then, a slight decrease occurred on the CF10. During the CR period, BHB significantly linearly declined and was restored to baseline on the CR3 (Figure 5a). At the same time, urine ketone also rose up to three “+” on the CF3 and top four “+” on the CF6 and CF9 in the routine urine test and recovered to a baseline of negative on the FR5 (data not shown). These results implied that an energy metabolic substrate shift from glucose to ketone occurred and a new balance formed during the middle of 10-day CF. As glucose oxidation finally generates 32 ATP and BHB to 22.5 ATP [30], the theoretically generated ATP was calculated with the formula ATP = 32 × C_glucose_ + 22.5 × C_BHB_. The results showed an increase of ATP from the CF6 and recovered to the BF on the FR5 (shown in Appendix A). Unexpectedly, HbA1c, an important index of average blood glucose, significantly declined on the CF3 and did not recover to baseline until the FR5 (Figure 5b). As a pivotal glucose sensor and regulator, insulin presented an immediate decline, achieved its lowest at 18.64 ± 11.67 pmol/L (Mean diff. −36.09 pmol/L, 95% CI (−55.28 to −16.90) on the CF3 and maintained at this low level during the fasting period. During the refeeding, insulin gradually rose and recovered to baseline on the FR5 (Figure 5c). From the calculated HOMA-IR, three participants were >2.0 and did not take any diabetes medication and with normal fasting glucose level, and most of the participants ranged from 0.9–2.0. All of them showed a rapid decrease on the CF3 and maintained their low level during the fasting period with the same trend as with insulin (Figure 5d). On the contrary, glucagon markedly increased by 13.56% (Mean diff. 16.21 pg/mL, 95% CI (0.98 to 33.39) on the CF3, 15.96% (Mean diff. 20.08 pg/mL, 95% CI (2.89 to 37.26) on the CF6 and 14.51% (Mean diff. 17.60 pg/mL, 95% CI (0.14 to 34.79) on the CF9, and fell to baseline in the following refeeding period (Figure 5e). However, there was no change occurring in glucagon-like peptide-1(GLP-1) during the whole experiment (Figure 5f).

The changes of lipometabolic indexes, including triglycerides (TG), cholesterol (CHOL), Low-density lipoprotein cholesterol (LDL-C) and high-density lipoprotein cholesterol (HDL-C) and leptin were ordinarily depicted in our previous reports [27]. In detail, there was a slight but not significant decrease of TG and HDL-C during the 10-day fasting period (Figure 6a,c). CF induced a significant increase of CHOL with 13.26% (Mean diff. 0.64 mmol/L, 95% CI [0.14 to 1.14] on the CF3, 29.33% (Mean diff. 1.42 mmol/L, 95% CI [0.79 to 2.05] on the CF6 and 33.17% (Mean diff. 1.61 mmol/L, 95% CI (0.99 to 2.24) on the CF9, respectively, then recovered to baseline after refeeding (Figure 6b). Moreover, the LDL-C showed an increase of 23.53% (Mean diff. 0.78 mmol/L, 95% CI (0.23 to 1.33), 45.39% (Mean diff. 1.55 mmol/L, 95% CI (1.05 to 2.06) and 43.63% (Mean diff. 1.48 mmol/L, 95% CI (0.87 to 2.08), respectively during the fasting period, and presented a gradual recovery to the baseline during eating resumption (Figure 6d). There was a marked elevation of leptin from 1.67 ± 0.55 ng/mL of the BF to 2.27 ± 0.86 ng/mL of the CF3 and the highest 2.48 ± 0.60 ng/mL of the CF9. Leptin showed a slight decrease after calorie restriction resumption to 2.06 ± 0.58 ng/mL and fully recovered to the baseline until the FR5 (Figure 6g). As the main carrier of HDL-C, ApoA1 declined by 11.43% (Mean diff. −13.19 mg/dL, 95% CI(−26.67 to 0.29)on the CF3, 14.97% (Mean diff. −17.18 mg/dL, 95% CI (−34.00 to −0.35) on the CF6 and 22.11% (Mean diff. −24.64 mg/dL, 95% CI (−39.28 to 10.00) on the CR3, then recovered to BF baseline on the FR5 (Figure 6e). On the contrary, ApoB, the main structural protein of LDL-C, significantly increased by 23.15% (Mean diff. 24.40 mg/dL, 95% CI (10.62 to 38.18) on the CF6 and 41.94% (Mean diff. 45.66 mg/dL, 95% CI (34.64 to 56.69) on the CF9, then recovered to the baseline on the CR3 (Figure 6f). Lipoprotein (a) presented a gradual increase from initial fasting (10.50 ± 8.59 mg/dL of the BF, 14.55 ± 8.96 mg/dL of the CF3, 16.78 ± 7.59 mg/dL of the CF6, 20.75 ± 10.55 mg/dL of theCF9 respectively) to CR3 (20.48 ± 12.31 mg/dL) and fell back to the baseline on the FR5 (9.56 ± 7.22 mg/dL) (Figure 6h). Nevertheless, no alterations of the adiponectin (Figure 6i) and lipoprotein lipase (LPL) (Figure 6j) occurred during the whole fasting experiment. It was noteworthy that lipase showed a gradual increase during the fasting period, but there was a significant elevation of 131.49% on the CR3 and 112.92% on the FR5 (Figure 6k). Carbondioxide-combining power (CO2-CP) measured by the PEPC enzyme method showed a significant gradual decline during the fasting period and quickly recovered to the baseline on the CR3 (Figure 6l).

### 3.6. Liver and Kidney Functions

Most of the liver function biochemical biomarkers did not change during the fasting and diet restoration, such as alanine aminotransferase (ALT), Aspartate aminotransferase (AST), alkaline phosphatase (ALP), Gamma-Glutamyl transferase (GGT) and total bile acid (TBA) (Figure 7a–e). There was a slight increase trend of TBA during the whole experiment and a notable decline of ALP on the FR5. It was worth noting that the ALT level of three participants and AST of one participant fleetly increased on the FR5, which were slightly higher than the upper threshold of the clinical standard, as was the TBA level.

As shown in Figure 7f, serum creatinine mildly increased during the fasting time until the CR3 (82.82 ± 13.92 μmol/L of the BF vs. 94.71 ± 17.01 μmol/L of the CR3). However, there was no significant change in urine creatinine during the whole experiment time, although a decrease of 38.08% was observed on the FR5 (Figure 7g). The estimated glomerular filtration rate (eGFR) showed a contrary change trend to the creatinine with a decrease of 7.52% on the CF6 and 12.05% on the CR3, respectively (Figure 7h). Most importantly, the level of cystatin C (Cys-C) markedly declined on the CF6 (23.99%) and CF9 (29.65%) when compared with that of the BF (Figure 7i). A slight increase (17.06%) of blood urea was observed on the CF3, which was smoothly recovered to the baseline following that time (Figure 7j). As a sensitive indicator of premature renal failure, a gradual increase of urine microalbumin (UMALB) was detected, especially on the CF9 (297.43%± 264.79%, *p* = 0.014) and the CR3 (409.91%± 636.71%, *p* = 0.042), but there was great individual difference (Figure 7k). The ratio of UMALB to urine creatinine almost did not markedly change with a stepped increase until the CR3 (Figure 7l). All the kidney function indexes were restored to the baseline level on the FR5.

### 3.7. Vitamins and Electrolyte Ions

Vitamins were required for various biochemical functions and should be supplied by diet. During the prolonged fasting, blood lipid-soluble vitamin (LSV) of vitamin A, E and D3 was significantly increased on the CF3 of 87.15% (Mean diff. 0.97 μmol/L, 95% CI (0.37 to 1.56), 17.77% (Mean diff. 1.94 μg/mL, 95% CI (0.44 to 3.44)and 233.09% (Mean diff. 93.31 μmol/L, 95% CI (62.53 to 124.10), respectively, and then maintained at their higher level during the fasting period and then recovered to prefasting level after diet (Figure 8a–c). No marked changes were observed in the water-soluble vitamins (WSV) as shown in Figure 8d–i. As expected, the level of sodium ion (Na) markedly reduced 2.88% (Mean diff.- 0.18 mmol/L, 95% CI (−1.92 to 1.57)) from the CF6, then under the lower limit of 137 mmol/L on the CF9 and restored to the baseline level on the FR5 (Figure 8j). The chlorine ion (Cl) level presented the same change trend as Na (Figure 8k). There were volatility changes in the potassium (K), calcium (Ca) and phosphorus (*p*) levels. Magnesium decreased about 7% from the CF6 to CR3 (Figure 8l–o).

### 3.8. Resting Metabolic Rate and Respiratory Quotient

To explore the energy metabolism during prolonged fasting, the resting metabolic rate (RMR) and respiratory quotient (RQ) were measured using the method of indirect calorimetry. As shown in Figure 9, RQ value grew closer to 0.7 with a fasting time extension from 0.833 ± 0.073 (BF1/2) to 0.716 ± 0.041 (CF9/10), then gradually increased to 0.765 ± 0.060 (CR3/4) and 0.868 ± 0.091 (FR4/5). These data implied that the energy metabolism switched to fat oxygenolysis as a prepotent pattern during the prolonged fasting. The calculated RMR showed an increase of 8.2% (Mean diff. 138.2 Kcal/day, 95% CI (44.25 to 320.6) on the CF3, and decreased about 5.3% (Mean diff. −110.4 Kcal/day, 95% CI (−304.80 to 84.04) on the CF9, 8.2% (Mean diff. −166.8 Kcal/day, 95% CI (−410.9 to 77.25) on the CR3 and 5.4% (Mean diff. −119.8 Kcal/day, 95% CI (−315.6 to 75.94) on the FR4. The maximum decrease could reach 25.5% on the CF9, 37% on the CR3 and 22.8% on the FR4. When compared with CF3, the RMR showed a marked decrease. Fasting assuredly reduced the resting energy metabolism and had strong individual differences.

## 4. Discussion

Increasing data from human and animal experiments demonstrated CR and fasting were associated with deceleration or prevention of most chronic metabolic illnesses and inflammatory diseases. The different and individual benefits implied that a suitable fasting regimen with a modulating diet and meal frequency as well as fasting period and interval time could represent a new paradigm for alleviating metabolic dysfunction, which should be further evaluated in controlled clinical trials and observational studies [19,31]. Most studies supported that fasting or CR improved systemic metabolic function, but whether prolonged fasting led to a new and stable metabolic homeostasis was largely unknown. The primary objective of this study was to examine the metabolic response and the potential changes in body composition, circulating hormones and metabolism regulating molecules to a prolonged water-only fasting in male adults.

**Tolerance and safety of prolonged fasting.** Because of the habit, physiology and mentality, long-term fasting for more than 36 h does not come easy for most people. In contrast, prevention and therapy of chronic metabolic disorders should take a long time. A suitable pattern with an appreciative interval time of fasting and repeated times is important, especially the tolerance from physiology and psychology. From our results, all participants completed the whole fasting experiment without any complaints and major side effects, which indicated that adults could well tolerate 10-day water-only deliberate fasting. Toledo reported that Buchinger periodic fasting lasting from 4 to 21 days was safe and well-tolerated [32], as did some other prolonged fasting human experiments from 5 days to 2 weeks to prevent obesity, hypertension [33], DM, insulin resistance [34] and arthritis [35]. Ketoacidosis is another pivotal problem during prolonged fasting. Our results showed the mean highest concentration of BHB was 5.616 ± 0.911 mmol/L on the CF8 and only two values were more than 7 mmol/L. Although there was a continuous decline of CO_2_-cp, the mean concentration was more than the lower limit threshold of 20 mmol/L. In total, 4 or 6 out of 13 participants showed lower than this threshold on the CF6 and CF9, which means there was a potential slight ketoacidosis risk. By the way, almost all blood routine indexes, TP, ALB and GLB were steady and there were no significant clinical changes. The increased tendency of blood and urine creatinine suggested that limited muscle protein was catabolized during the CF. Previous reports and present data suggested that an initiative long-term fasting for more than 5 days was a safe feasible model and could be endured by human beings for health or therapy goals.

**The metabolic switch and homeostatic models during complete fasting.** Except for the safety concern, another focus on prolonged fasting was the metabolic homeostasis and energy substrate shift. We measured blood glucose and BHB every day during 10-day CF. A new homeostasis of energy source formed between 3–6 days of CF, as the lowest level of glucose on the CF4 and the highest level of BHB on the CF5 in the present experiment (Figure 5a). This new dynamic balance kept steadily in the subsequent fasting time, documenting the metabolic switch. These results were consistent with previous reports that ketosis had been shown to reach a plateau after 4–5 days of prolonged CF [36,37]. The increased ATP calculated from glucose and BHB suggested fasting did not lead to energy deficiency even during the complete food deprivation. Additionally, blood glucose stabilized at the lower normal level, which implied a systemic homeostasis of gluconeogenesis. Moreover, there was no alteration of TG (Figure 6a) during fasting and refeeding, which meant most of TG released from accelerated lipolysis contributed to gluconeogenesis as a previous report [38]. The individual difference of TG level showed by SD value become smaller during fasting, which predicted a necessary level of TG to body. A slight increase of serum urea (Figure 7j) suggested the gluconeogenesis contributed by amino acid was minor or tightly regulated to avoid protein excessive catabolism [39], as consistent with previous reports [40,41]. The hypoglycemic insulin (Figure 5c) and the hyperglycemic glucagon (Figure 5e) reached their lower (18.64 ± 11.67 pmol/L, −65.45%) or highest (155.78 ± 25.32 pg/mL, +15.95%) platform on the CF3 or CF6, respectively, and maintained their level in the subsequent fasting period. These two antagonistic glucose-regulating hormones also reached new homeostasis. Following the decrease of blood glucose, a significant decline of HbA1c and HOMA-IR was observed throughout the fasting experiment (Figure 5b,d), whereas one-week Buchinger fasting did not significantly decrease HbA1c and HOMA-index in a pilot study of type 2 diabetes patient [42]. Moreover, the multifaceted GLP-1 was unaffected during the fasting experiment. The new homeostasis was also supported by the similar declined degree of RMR with the BW decrease except for the first 3 days. From an evolutionary point, animals instinctively consumed more energy to find food in the initial stage of food deprivation. These data implied a new and stable energy balance after 3-day fasting between BW, an alternate energy substrate and physiological energy consummation [41].

**The complex alterations of lipid metabolism**. Lipolysis plays a pivotal role as an energy source and is precisely regulated to survive longer during food deprivation. From the anthropometric analysis, the BW speedily declined in the initial days of fasting to support the increased RMR. RQ closed to 0.7 from the CF3 to CF9 indicated the depletion of glycogen and major expenditure of fat after 3-day fasting. Body composition analysis showed that the fat mass continuously lost in the whole experimental period which mainly occurred in limbs and trunk, especially in subcutaneous and visceral fat supported by the more decrease of SFT and EVM (Figure 4n,k). Additionally, the declined lean mass or fat-free mass recovered after 4-day CR and 5-day FR. These results implied that the body was mainly a nutritional supplement, not gained weight in the initial refeeding days. In the lipid metabolic indexes, it was reported that IF decreased the total cholesterol by about 10%–21%, TG by about 14%–42% and LDL-C [20], but not all [43]. In the present study, total cholesterol and LDL-C increased during fasting and recovered to lower after refeeding. The early report showed that fasting for 1 week resulted in a significant elevation of serum cholesterol and TG by 25% and continued fasting for up to 21 days resulted in lowering of both cholesterol and TG to pre-fast levels [44]. There are complicated alterations of these lipid metabolic indexes observed after long-term Buchinger fasting [42,45,46] in which participants were allowed to intake 200–250 Kcal daily such as tea or honey. CF resulted in an increase of TC, LDL-C and Apo-B [47]. It is well known that LDL with ApoB mainly brings cholesterol from the liver to peripheral tissues and HDL plays a reverse role in cholesterol transport with ApoA1. They presented the same tendency during this prolonged fasting, and so did the lipoprotein, which was dependent on the degree of fasting [48]. Most reports showed fasting decreased the leptin level which played important roles in the regulation of thyroid and metabolism [49]. However, our results showed an increase of leptin during CF, which should be investigated later. These results indicated the main energy source of lipid was tightly regulated for extending the living period during prolonged food deprivation. Taken together, lipid metabolism presents different changes model between IF and prolonged CF, which may result from the initial state or fasting degree. Blood lipids play a causal role in the etiology of metabolic diseases. The distinct changes from different fasting regimens increase the clinical application difficulty of fasting for therapeutic purpose.

**The interesting changes in physiological and nutritional status.** First, fasting had a positive effect on blood SBP as in previous reports, but almost no change of DBP [50,51]. SBP did not decrease in the short fasting group (<7 days), but occurred in the prolonged fasting (>7–30 days) [50], the same as our observation. Ramadan intermittent fasting had no effects on their hypertensive state [52] or decreased SBP and DBP [53]. On the contrary, pulse rate significantly increased from the CF2 and throughout the whole fasting and CR period [41], whereas Pratap observed a decrease of pulse in the prolonged fasting group [50]. Fasting could also prolong QTc intervals [51]. However, there were no alterations in the myocardial enzymogram (CK, CK-MB, LDH, HBDH, AST), which indicated prolonged CF caused no damage to cardiac tissue and the increased pulse may come from the energy metabolic shift in heart. A deep and systematic analysis should be done in the future on the cardiovascular system during prolonged fasting. Second, protein synthesis was inhibited during water-only prolonged fasting as shown by the marked decrease of short-life protein including prealbumin, RBP and transferrin, but not the key long-life protein that maintained the homeostasis of the humoral system such as TP, ALB and GLB. These results indicated that the body selectively synthesizes some vital proteins and liver function was not affected. Third, surprising results came from the changes of vitamins and electrolyte ions. In addition to the untested vitamin K, all other LSV were significantly elevated, whereas there were almost no changes in the WSV (Figure 8a–i). In regular views, fasting leads to nutritional deficiency, especially vitamins. Multivitamin was supplied daily in the early fasting experiments [54,55]. Recent evidence was consistent with our observation. Eight days of fasting induced statistically significant changes in the levels of three vitamin D metabolites with an increase of 24,25(OH)_2_D_3_ and 3-epi-25(OH)D_3_ and a decrease of 25(OH)D_2_ [56]. Aksungar reported Vitamin B12 and B9 showed a marked rise in the Ramadan fasting group [57], but only an increasing trend of vitamin B2 and a decrease trend of vitamin B6 occurred in our fasting experiment. It was worth noting a sharp decrease of vitamin B1 and B9 on the CF9. The LSV reached their highest plateau on the CF3 or CF6, which was still lower than their toxic level. These data implied the vitamins were released from stored tissues to maintain vitamin nutrition of blood and other tissues, especially the massive LSV following lipolysis. The changes of Na and Cl deserved the most attention in the aspect of electrolyte ions, because most participants’ blood Cl was closed to the lower threshold from the CF6 to CR3 and Na on the CF9. The changed degree was higher than the report from Toledo [32]. The concentration of other detected ions was within the normal ranges. It also should be considered in the decrease of TBW, ICF and ECF (Figure 4l) to maintain the electrolyte homeostasis. By the way, renal function showed some decline from the changes of eGFP, Cyc-C and UMALB, which were maintained within the reference values and returned to baseline after dieting as in previous reports [40].

## 5. Conclusions

Taken together, our results demonstrated that adults can well tolerate a 10-day CF in physiology and psychology with no major side-effects. The energy metabolic supply reached a new and stable dynamic balance between 3–6 days and body homeostasis could be maintained by accelerated fat decomposition, accompanied with reduced resting energy expenditure. Vitamin supplementation was not needed, but the NaCl should be considered. These findings could contribute to the development of a prolonged fasting therapeutic strategy with at least 3–6 days of CF to form new metabolic homeostasis. The limitations of the present study were the small number and only male participants, because of the risk control and condition constraints.

## Figures and Tables

**Figure 1 nutrients-14-03860-f001:**
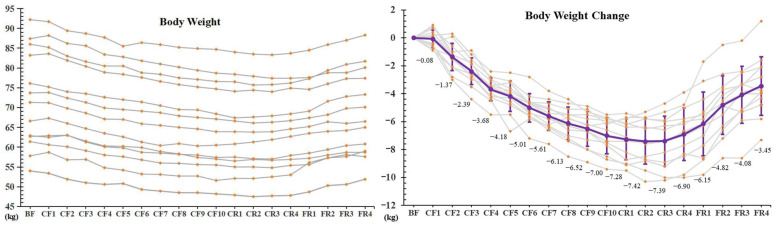
The alterations of body weight during 10-day CF and refeeding period. ** *p* < 0.01 vs. BF; # *p* < 0.05, ## *p* < 0.01, vs. CF10, *n* = 13. BF: before fasting, CF: complete fasting, CR: calorie restriction, FR: full recovery; the number depicts days.

**Figure 2 nutrients-14-03860-f002:**
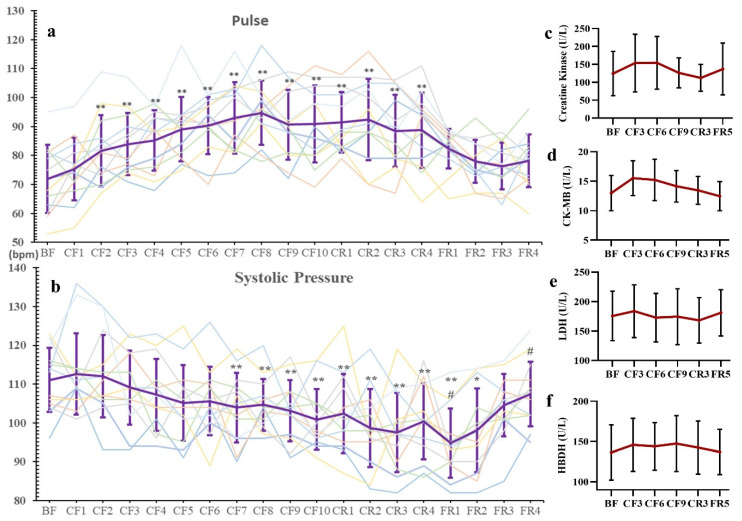
The changes of pulse (**a**), systolic pressure (**b**) and Myocardial Enzymograms (**c**–**f**) during 10-day CF and refeeding period. Every line in a and b present its change of one participant. * *p* < 0.05, ** *p* < 0.01 vs. BF; # *p* < 0.05, vs. CF10, *n* = 13. BF: before fasting, CF: complete fasting, CR: calorie restriction, FR: full recovery; the number depicts days. CK: creatine kinase, LDH: lactate dehydrogenase, HBDH: α-hydroxybutyric dehydrogenase.

**Figure 3 nutrients-14-03860-f003:**
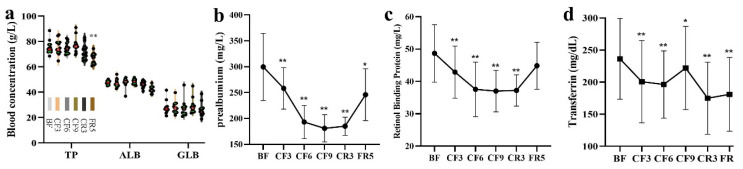
The changes of nutrition protein indicators during 10-day CF and refeeding period. (**a**) the blood concentration of nutritional protein indicators; (**b**) prealbumin (**c**) retinol binding protein (**d**) transferrin. * *p* < 0.05, ** *p* < 0.01, vs. BF; *n* = 13. BF: before fasting, CF: complete fasting, CR: calorie restriction, FR: full recovery; the number depicts days. TP: total protein, ALB: albumin, GLB: globin.

**Figure 4 nutrients-14-03860-f004:**
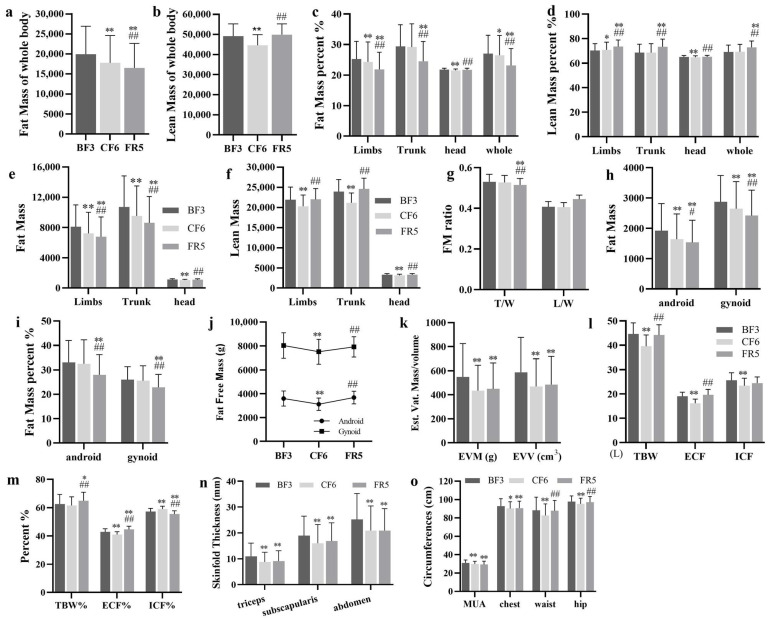
The body composition alterations of the whole and compartments were estimated by DEXA (**a**–**k**) or sfb7 (**l**,**m**) during 10-day CF and refeeding period. Skinfold thickness (**n**) and circumference (**o**). * *p* < 0.05, ** *p* < 0.01 vs. BF; # *p* < 0.05, ## *p* < 0.01, vs. CF6, *n* = 13. BF: before fasting, CF: complete fasting, CR: calorie restriction, FR: full recovery; the number depict days. T/W: Trunk/Whole, L/W: Limbs/Whole, EVM: estimated visceral adipose tissue (EST.VAT) mass, EVV: estimated visceral adipose tissue volume, TBW: total body water, ECF: extracellular fluid, ICF: intracellular fluid, MUA: mid-upper arm.

**Figure 5 nutrients-14-03860-f005:**
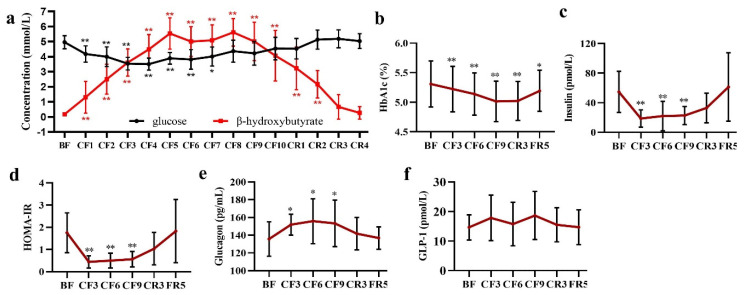
The alteration of glucose (**a**,**b**) and glycoregulators (**c**–**f**) during 10-day CF and refeeding period. * *p* < 0.05, ** *p* < 0.01 vs. BF; *n* = 13. BF: before fasting, CF: complete fasting, CR: calorie restriction, FR: full recovery; the number depicts days. HbA1c: Hemoglobin A1c, HOMA-IR: Homeostatic Model Assessment of Insulin Resistance; GLP-1: glucagon-like peptide-1.

**Figure 6 nutrients-14-03860-f006:**
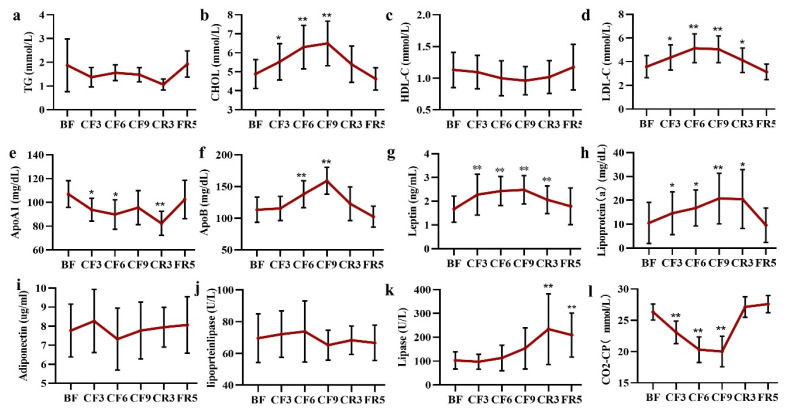
The changes of lipometabolic indices during 10-day CF and refeeding period. (**a**–**l**) represent the Y-axis coordinates. * *p* < 0.05, ** *p* < 0.01 vs. BF; *n* = 13. BF: before fasting, CF: complete fasting, CR: calorie restriction, FR: full recovery; the number depicts days.TG: triglycerides, CHOL: cholesterol, LDL-C: Low-density lipoprotein cholesterol, HDL-C: high-density lipoprotein cholesterol, ApoA1: apolipoprotein A1, ApoB: apolipoprotein B, CO2-CP: carbondioxide combining power.

**Figure 7 nutrients-14-03860-f007:**
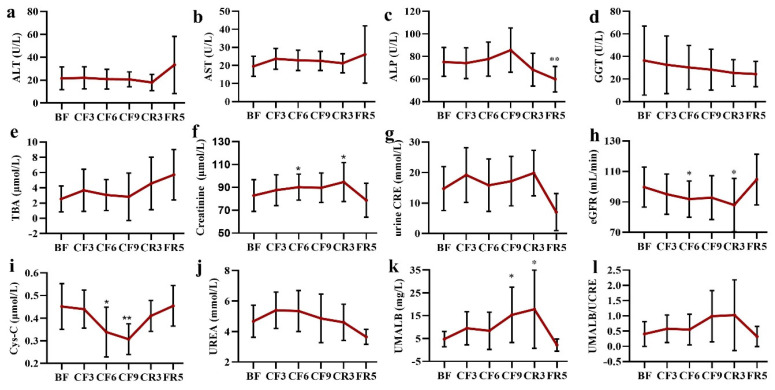
The alteration of liver (**a**–**e**) and kidney (**f**–**l**) function indexes during 10-day complete fasting experiment. * *p* < 0.05, ** *p* < 0.01 vs. BF; *n* = 13. BF: before fasting, CF: complete fasting, CR: calorie restriction, FR: fully recovery, the number presents days. ALT: alanine aminotransferase, AST: aspartate aminotransferase, ALP: alkaline phosphatase, GGT: Gamma-Glutamyl transferase, TBA: total bile acid, Cys-C: cystatin C, UMALB: Urine microalbumin, CRE: creatinine, eGFR: estimated glomerular filtration rate.

**Figure 8 nutrients-14-03860-f008:**
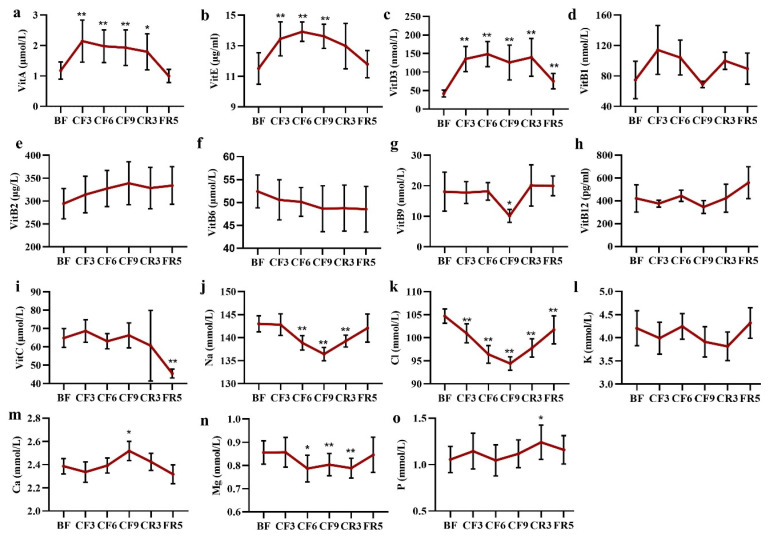
The alteration of main vitamin (**a**–**i**) and electrolyte ions (**j**–**o**) during 10-day CF and refeeding period. * *p* < 0.05, ** *p* < 0.01 vs. BF; *n* = 13. BF: before fasting, CF: complete fasting, CR: calorie restriction, FR: ful recovery; the number depicts days. Vit: vitamin.

**Figure 9 nutrients-14-03860-f009:**
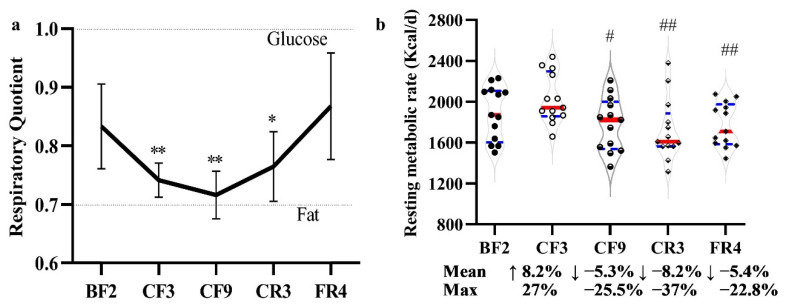
The effect of 10-day CF on the respiratory quotient (**a**) and resting metabolic rate (**b**). * *p* < 0.05, ** *p* < 0.01 vs. BF; # *p* < 0.05, ## *p* < 0.01 vs. BF; *n* = 13.BF: before fasting, CF: complete fasting, CR: calorie restriction, FR: full recovery; the number depicts days.

**Table 1 nutrients-14-03860-t001:** The changes of blood routine parameters.

	BF	CF3	CF6	CF9	CR3	FR5
MONO%	8.16 ± 1.67	9.53 ± 2.38	9.85 ± 2.86	9.90 ± 2.39 *	8.88 ± 1.54	9.37 ± 1.70
MONO (10^9^/L)	0.47 ± 0.10	0.61 ± 0.19 *	0.58 ± 0.19	0.57 ± 0.19	0.45 ± 0.12	0.49 ± 0.12
NEUT (10^9^/L)	3.27 ± 1.20	3.85 ± 1.73	3.33 ± 1.12	3.24 ± 1.65 *	2.44 ± 0.82 *	2.70 ± 1.02
NEUT%	53.60 ± 5.92	57.15 ± 6.78	54.83 ± 6.22	51.54 ± 9.25 *	47.78 ± 7.97 *	50.21 ± 7.28
RBC (10^12^/L)	4.77 ± 0.57	4.78 ± 0.58	5.03 ± 0.61	5.39 ± 0.63 *	5.08 ± 0.60	4.61 ± 0.49
HGB (g/L)	147.92 ± 14.36	148.69 ± 15.30	156.62 ± 15.60	160.46 ± 14.65 *	153.08 ± 12.66	137.77 ± 13.75
HCT (%)	43.22 ± 3.51	43.35 ± 3.56	45.28 ± 3.49	47.30 ± 3.40 **	44.10 ± 2.67	42.25 ± 3.39

* *p* < 0.05, ** *p* <0.01, vs. BF *n* = 13. BF: before fasting, CF: complete fasting, CR: calorie restriction, FR: full recovery; the number depicts days.

## Data Availability

The data presented in this study are available on request from the corresponding author.

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
