# Peer review of "Effects of 10-Day Complete Fasting on Physiological Homeostasis, Nutrition and Health Markers in Male Adults"

_nutrients, 2022, doi:10.3390/nu14183860_

Round 1
Reviewer 1 Report
The authors described effects of a 10-day fasting period on several outcome measures in healthy male adults. The novelty of this work lies in the wide range of outcome measures analyzed. However, none of the observed effects is original, which is definitely a limiting factor. Another limitation of this work is the descriptive and wordy way the results are presented and discussed; mechanistic insights are rare and the interpretation of the results is not always based on statistics, e.g. : "...small but insignificant increase tendency of blood and urine creatinine suggest ...". Moreover, the manuscript text needs extensive language editing, including headings, e.g. "Prolonged fasting can be well tolerated and safety" and captions. That said, the manuscript cannot be recommended for publication.
Author Response
Reply: Thank you for reviewing our manuscript and giving some valuable and helpful suggestions, which is highly appreciated. As your mentions, we systemically evaluated the safety, physiological and nutritional changes during the 10-day complete fasting. To our known, it is the first time to thoroughly describe the scheduled effects of prolonged complete fasting with the wide range of outcomes. As you known, fasting had been practiced for millennia and growing reports demonstrated its potential benefits for health. At present, most of the prevalent IF, ADF and TRF (Ramadan) refer to complete fasting lengths from 16 h to 48 h, or circulate between fasting (no more than 48 h) and refeeding. The metabolic switch may not truly take effects during short time complete fasting or calorie restriction because the glycogen, free fatty acid and amino acid stored in the tissues were able to last for 24h to 36h. Short time complete fasting may repeatedly deposit and mobilize these molecules. We proposed that the short time of complete fasting was the main cause of different or individual benefits from fasting. Moreover, little is known about the safety and physiological changes of long-term fasting, which lead to the worries of prolonged fasting and still a lack of a safe fasting protocol to guide physicians in its prescription as Rita and Francesca reported (Nutrients. 2021; doi: 10.3390/nu13051570). To highlight this view, we modified the introduction section (paragraph 2 and 4) and added these sentences to detailly descript our hypothesis. We expect this explanation could help you better understand our work.
We agreed with the second limitation you mentioned. Thanks very much. We tried to present a detailed alternations of the body during prolonged fasting. Because there are many measured parameters to be given expression, the results section manifest in the description and way. But in the discussion section, we summarized the overall change trend from safety, metabolic homeostasis, lipid metabolism, physiological and nutritional status. In the revised manuscript, we edited some redundant words and sentences, and gave some mechanistic explanation.
By the way, we also changed some describe way of sentence with detailed statistic results. Some subtitles, grammar and tense issues were also corrected in the revised manuscript by English expert. We sincerely hope that this revised manuscript make you better understand our work. Thanks again.
Reviewer 2 Report
The purpose of this study was to investigate the safety, time effects of metabolic homeostasis and healthy indexes during prolonged fasting. The authors concluded that the liver function was not influenced, but a slight decrease of kidney function presented. The interesting results came from the marked increase of lipid-soluble vitamins and significant decrease of sodium and chlorine. Adults could well tolerate a 10-day fasting. The new metabolic homeostasis was achieved. No vitamins but NaCl supplement should be considered. These findings provide evidences to design a new fasting strategy for clinical practice.
The manuscript is well written and present interesting findings. However some modifications are required:
I suggest that some studies related to Ramadan Intermittent fasting could be helpful for the rational of the study.
I suggest adding some hypothesis at the end of the introduction.
L.112. “Three” instead of “three”.
Do the authors calculate the required sample size.
Statistical analysis:
- Do the authors verify the normality of the disruption
- I suggest adding the effect size and the confidence interval
I suggest adding and discussing the limitation of the study at the end of the discussion.
Reviewer 3 Report
Zhongquan Day and colleagues have described the effect of 10 days of fasting (only water ad libitum) in thirteen male patients followed by a period of caloric restriction and a final stage of complete recovery. Thanks to this study, the authors have demonstrated the safety of a complete 10-day fast by evaluating different parameters such as markers of heart, liver and kidney function, lipid profile, blood count, body composition, macro and micronutrients in plasma, etc. At different points of the nutritional trial. The authors conclude that this protocol of 10 days of complete fasting has no adverse effects, and that only NaCl supplementation should be considered. The study is novel and well designed. However there are some issues that need to be resolved:
Minor comments:
- In Figure 8c, the y-axis legend is different from the rest of the panels.
- Line 543. There is a mistake in the first word, the complete word appears as “ccomplete”.
- The analysis of figure 5a is not described in the manuscript. In the “Statistical Analysis” section, only the repeated measures ANOVA is described, and the correlation analysis is not described.
Major comments:
- In the title of this article "Effects of 10-day complete fasting on the physiological homoestasis, nutrition and health markers in normal male adults" the word "normal" is included. What does normal mean in this context? Does this refer to healthy? If it refers to healthy, the patients participating in the study had a BMI between 19.2-31.9 (lines 210-211), the patients ranged from normal weight to class I obesity, so healthy or normal should not be considered. This point needs to be explained.
- Regarding the recruitment of volunteers, why was it only considered to include men in the study? It is difficult to reach a conclusion of a nutritional intervention for therapeutic purposes, when 50% of the population is excluded from the study.
- It has been described that there are differences between different races of the human population regarding glucose metabolism and its implication in fasting (DOI:10.2337/dc10-0898). It would be interesting to describe which population is in the study. In order to explain possible differences with other present and future studies.
- Regarding the exclusion criteria, the authors have described different pathologies that are considered exclusion criteria. However, nothing is mentioned about insulin resistance. Did these patients have any degree of insulin resistance? According to the HOMA-IR, a value below 1 is considered optimal insulin sensitivity. However, in Figure 5e, the HOMA-IR value at point BF is >1, reaching values >1.9 in some cases. This would indicate early insulin resistance in some of the participants. Could different results be observed if the participants are sensitive or have some degree of resistance to insulin? This point needs to be explained.
- Regarding the measurement of blood pressure and heart rate, the authors describe that the measurements were made immediately after they woke up. Can the authors give the specific time range? Did the participants drink any liquid prior to these measurements? Or did they show some degree of dehydration at the time of the measurement? There is a circadian pattern in these patterns (DOI:10.1016/j.hrthm.2018.08.026) and hydration status (10.3390/nu11081866).
- Some errors appear in the table in figure 3. For example, in the text (lines 250-254) it is indicated that there are significant differences in CF9 in RBC, HGB and HCT. However, these significant differences appear not indicated in the table. The statistics of the table should be reviewed in depth. All existing differences vs. BF should be included.
- In line 255. It is indicated that there are no changes in the other 17 parameters evaluated. What parameters are those? It would be very helpful to include all this data for these parameters in a supplementary table.
- On lines 284-288. It is described that FM% decreases, but does this decrease refer to a global drop in FM% or in the different parts of the body analyzed? If you are referring to a reduction in overall FM%, it would be helpful to include a graph in Figure 4 showing this result.
- It would be helpful if the individual values were plotted on a graph, in order to see the individual data (Figure 3a and 4).
- In line 388, the authors indicate that there is a marked increase of GLP-1 in CF9. However, this difference is not indicated in the graph (Figure 5g).
- Figure 6g shows how during fasting (from CF3 to CF9) there is a significant increase in plasma leptin levels, however during this same fasting (CF6) there is a reduction in all fatty depots (Fat mass of whole body, Fat mass percent% of limbs, head and whole, etc). Should the authors explain how the amount of adipose tissue is reduced but plasma leptin levels are increased? On the other hand, leptin is an anorexigenic hormone, increasing satiety signals. This point should be well explained by the authors.
- In figure 6i, it is shown that adiponectin does not change during the nutritional test. However, different studies show a variation in adiponectin levels due to different types of caloric restriction and body fat loss (DOI:10.1016/j.clnu.2020.10.034). On the other hand, the determination of adiponectin is complex. What method has been used to determine adiponectin? If the answer is ELISA technique, what isoform has been detected, globular adiponectin, trimer, hexamer, multimer? This information is relevant and may explain the absence of differences on adiponectin. The authors should detail in the material and methods section and discuss this point in depth.
- The HOMA-IR is an indicator of insulin sensitivity, but in recent years the leptin/adiponectin ratio has been described as a good marker of insulin sensitivity. Have the authors seen any difference in this ratio?
- During the study, the authors have observed that fasting from CF3 increases cholesterol levels, LDL-C and ApoB, above normal levels, even reaching pathological values. Although the authors make a small comment about this in lines 605-608, the authors should discuss this point further. On the other hand, other studies (DOI:10.1371/journal.pone.0209353) observe a reduction in the levels of LDL-C and other lipids in similar nutritional interventions. These discrepancies should be discussed by the authors.
- In Figure 8, the authors observe a decrease in Na and Cl ions, even reaching out of the healthy range. However, other authors have not reported this result. For example Doi: 10.1172/JCI83349; 10.1210/jcem-60-6-1120 and 10.1172/JCI118137) propose KCl supplementation due to the risk of hypokalemia, not hyponatremia. Possible differences should be discussed. Was water intake monitored during the study? Is it possible that they have replaced the lack of food with an excessive intake of water?
- In lines 485-494 the authors first describe the effect of the nutritional study on the Respiratory Quotient (RQ) and then the Resting metabolic Rate (RMR). However, in the graphs they appear in the opposite order. It would make it easier to follow the order of the text to place the figures. Also, it is described in the text that there is a significant increase in RMR but it does not appear in the figures. What are these discrepancies between text/figures due to?
- In lines 376-379 the authors describe an increase in ATP during fasting, calculated from glucose and BHB. It would be very interesting to include a panel in Figure 5 showing these results.
- In Figure 6 the authors show the effect of the nutritional intervention in a very complete panel of the lipid profile. However, non-esterified fatty acids (NEFAs) do not appear in the graphs. These NEFAs provide a lot of information about the metabolic status of patients. Please include a panel with these measurements if possible.
Round 2
Reviewer 1 Report
The authors tried to address my previous comments by including statistics and having the manuscript langage edited. However, the level of originality is still relatively low - with respect to both, the overall hypothesis and the results presented. It is well-known, for example, that blood glucose measurements by strips and enzymatic accsays correlate well. That said, the two (!) graphs devoted to this message should be deleted. The presentation of the results and the discussion are still wordy and not concise, and the entire manuscript text still needs language editing in order to convey the author's messag, e.g. "The FM percentage of these two parts were no significant alternation on the CF6" .
Reviewer 2 Report
no further comments
Author Response
Thank you very much.
Reviewer 3 Report
Minor comments:
- In the legend of table 1 the letter p of the word parameters is missing.
- Review all legends of all figures and tables. After numbering the figure, begin the sentence with a capital letter. E.g "Table 1. the changes of blood...." should be "Table 1. The changes of blood...."
- Check the font size of the entire manuscript, it must present the same font size and style. E.g. Line 140-141 doesn't seem to have the same font size as the rest of the text.
Author Response
- In the legend of table 1 the letter p of the word parameters is missing.
Response: Thanks very much for your careful review and helpful suggestion. We added the missing alphabet.
- Review all legends of all figures and tables. After numbering the figure, begin the sentence with a capital letter. E.g "Table 1. the changes of blood...." should be "Table 1. The changes of blood...."
Response: Thanks very much for your benignant suggestion. We checked all the legends and fixed them.
- Check the font size of the entire manuscript, it must present the same font size and style. E.g. Line 140-141 doesn't seem to have the same font size as the rest of the text.
Response: We quite appreciate your favorite advice. We careful check all the manuscript to keep the same font size following the Journal’s format.